# JOINT IMPORTANCE SAMPLING FOR VARIATIONAL INFERENCE

**Jack Klys**[1]**, Jesse Bettencourt**[2] **& David Duvenaud**[2]
[1]Department of Mathematics
[2]Department of Computer Science
University of Toronto
{jessebett, duvenaud}@cs.toronto.edu
{jack.klys}@mail.utoronto.ca

## ABSTRACT

We consider methods of variance reduction in Monte Carlo estimators which arise from importance sampling, with application to variational inference. We show that learning dependencies between samples, while preserving their marginal distributions outperforms sampling techniques which assume independence among samples in some settings.

## 1 INTRODUCTION

Let $p(x, h)$ be the probability distribution of a latent variable model on a space $X \times H$. When the marginal likelihood of the data $p(x)$ is intractable it can be estimated by constructing an unbiased estimator $\hat{p}$ for $p(x)$ defined on some measure space $(Z, \Omega, \mu)$. The model can then be optimized by maximizing the evidence lower bound (ELBO):

$$\log p(x) = \log E_\mu(\hat{p}) \geq E_\mu \log(\hat{p}) = \mathcal{L}(\hat{p}). \tag{1}$$

Suppose we wish to estimate a quantity $x$ by Monte Carlo. Let $Y_1, \ldots, Y_k$ be random variables on a measure space $(Z, \Omega, \mu)$ and $f_1, \ldots, f_k$ real-valued functions such that $f_i(Y_i)$ is an unbiased estimator of $x$, so $E_\mu(f_i(Y_i)) = x$. Then we can form the unbiased estimator of $x$

$$\theta = \frac{1}{k} \sum_i f_i(Y_i). \tag{2}$$

It is a classical problem to determine $f_i$ and $Y_i$ such that $\theta$ has minimal variance (Wilson, 1983).

This type of estimator appears in the training of generative models. When an approximate posterior distribution $q(h \mid x)$ is trained alongside $p(x, h)$ we can let $f_i(h) = p(x, h)/q(h \mid x)$ and $Y_i \sim q(\cdot \mid x)$ to obtain the estimator

$$\hat{p}_k = \frac{1}{k} \sum_i \frac{p(x, h_i)}{q(h_i \mid x)} \tag{3}$$

a process refered to as importance sampling. The variance of this estimator goes to 0 with $\mathrm{KL}(p(h|x)||q(h|x))$. This was the estimator used in Burda et al. (2015) to train Importance Weighted Autoencoders (IWAE).

## 2 VARIANCE OF AN UNBIASED ESTIMATOR AND THE ELBO

We start by making precise the connection between the variance of an estimator and the ELBO. Chebyshev's inequality states that we have, for any $\epsilon > 0$

$$\mu(|\hat{\theta} - \theta| \geq \epsilon) \leq \frac{\mathrm{Var}(\hat{\theta})}{\epsilon^2}.$$

Using this we prove

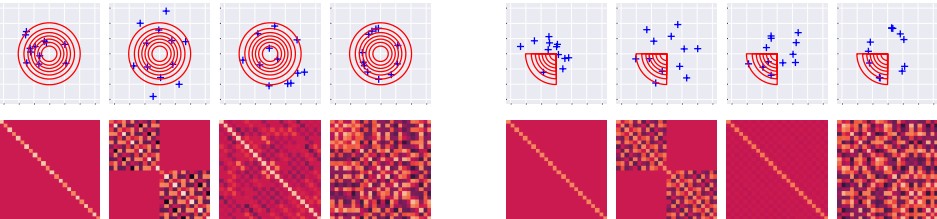

Figure 1: The result of maximizing $\mathcal{L}(\hat{p}_M)$ for two fixed target distributions, plotted in red. The joint distributions (defined in Section (3)) used are, from left to right, $q_{\text{IWAE}}$, $q_{\text{COVRn}}$, $q_{\text{COVK}}$, $q_{\text{COVD}}$. Top: one set of samples drawn from each model. Bottom: The covariance matrix of the corresponding model.

**Theorem 2.1.** *Let $\hat{\theta}$ be an unbiased estimator of $\theta$. There exists a constant $C$ (independent of $\hat{\theta}$) such that for any sufficiently small $\epsilon > 0$ (depending only on $\theta$)*

$$\log \theta - \mathcal{L}(\hat{\theta}) < C \left( \epsilon + \frac{\text{Var}(\hat{\theta})}{\epsilon^2} \right).$$

In particular this implies that if the sequence $\{\hat{\theta}_i\}$ satisfies $\text{Var}(\hat{\theta}_i) \longrightarrow 0$ then $|\theta - \mathcal{L}(\hat{\theta}_i)| \longrightarrow 0$.

In Maddison et al. (2017) a similar result is proven which expresses $|\theta - \mathcal{L}(\hat{\theta}_i)|$ in terms of the variance and higher moments of $\hat{\theta}$. We refer to the appendix for the proof of Theorem 2.1, which generalizes theirs, and a remark about the relationship between these two results.

As an example, from (3) it is easy to see that $\text{Var}(\hat{p}_k) = \text{Var}(\hat{p}_1)/k$ if the $Y_i$ are independent. Henceforth we will refer to this estimator as $\hat{p}_{IWAE}$ and omit the subscript $k$. We will define some more examples of unbiased estimators and show that they have lower variance than $\hat{p}_{IWAE}$.

## 2.1 REDUCING THE VARIANCE

Motivated by the above discussion we look at the problem of variance reduction in estimators of the form (2). Note that in (3) all the $Y_i$ are independent and equal to $q(h \mid x)$. However in general we have $\text{Var}(\theta) = \sum_{i=1}^{k} \text{Var} f(Y_i) + 2 \sum_{i<j} \text{Cov}(f(Y_i), f(Y_j))$. Hence by choosing appropriate dependencies between the $Y_i$ we can make the covariance term on the right negative, thus lowering the variance. This is equivalent to finding a distribution on $Z^k$ with marginals $f(Y_i)$ and non-diagonal covariance matrix.

We apply this idea to estimators in the form of (3). That is $f_i(h) = p(x, h)/q(h \mid x)$ but $Y_i \sim q(\cdot \mid x)$ is the marginal of a distribution $q(h_1, \ldots, h_k)$ on $Z^k$. We will denote this estimator by $\hat{p}_M$.

In addition to the usual parameters of the marginals $q(h_i)$ we parametrize a covariance matrix between them. We then maximize $\mathcal{L}(\hat{p}_M)$ with respect to these parameters, both in the setting when $p$ is a fixed target distribution, as well as when it is defined by some parameters, as with a VAE. Figure (1) demonstrates the result of this for two fixed targets $p$.

In Cremer et al. (2017) it is shown that maximizing $\mathcal{L}(\hat{p}_M)$ with respect to the parameters of $q(h_i)$ is equivalent to minimizing $\text{KL}(p(h|x)||q_{\text{EW}}(h))$ (for the definition of $q_{\text{EW}}$ see Section 4.5). Though it is discussed in the case when the marginals are independent it holds in our setting as well. Thus by using estimators capable of achieving lower variance we expect to better fit $q_{\text{EW}}$ to the target posterior $p(h \mid x)$.

## 2.2 OPTIMIZING THE SAMPLER

The problem of minimizing the variance of (2) has been studied classically. Theorem 1 from Wilson (1983) shows that given fixed functions $f_i$ with $i = 1, \ldots, k$ one can get arbitrarily close to the minimal possible variance by taking a uniform random variable $U_0 \sim U(0, 1)$ and choosing appropriate

| | Mixture | | Half | | Slice | | MNIST |
|---|---|---|---|---|---|---|---|
| | 2d | 10d | 2d | 10d | 2d | 10d | |
| IWAE | 0.179 | 6.383 | 0.046 | 1.088 | 0.179 | 0.426 | **87.544** |
| IWAEN | 0.118 | 5.883 | **0.000** | 1.018 | 0.118 | 0.367 | 87.754 |
| COVR6 | **0.027** | 10.091 | **0.000** | 1.079 | **0.027** | 0.435 | 87.900 |
| COVK | 0.094 | 9.450 | 0.011 | 0.974 | 0.094 | 0.310 | 87.592 |
| COVD | 0.054 | **1.277** | 0.005 | **0.899** | 0.054 | **0.229** | - |

Table 1: Negative log-likelihoods when fitting to different targets. The last column is the result of training a VAE (here we report the minimum the model achieved during training). Models with non-diagonal covariance matrices give improvement in low dimensions, but show little difference with a VAE.

transformations $T_i$ $i = 1, \ldots, k$ satisfying certain regularity conditions, see Wilson (1983), and setting $Y_i = T_i(U_0)$. That is, only one sample is taken, and the rest are obtained from it deterministically. The proof is however non-constructive and it is not clear in general what the transformations $T_i$, and hence the covariance matrix of $q$ should be.

The marginals of $\hat{p}_M$ are normal distributions on $\mathbb{R}^d$ so the covariance matrix of $q(h_1, \ldots, h_k)$ is a $kd \times kd$ matrix, where the diagonal $d \times d$ blocks are the covariance matrices of the marginals. Thus we want to determine an optimal covariance matrix for $q$.

## 3 EXPERIMENTS

We first demonstrate the effect of learning a normal distribution $q$ on $\mathbb{R}^{dk}$ by minimizing $-\mathcal{L}(\hat{p}_M)$ for a fixed target distribution $p$.

Let $q = N(\vec{\mu}, \Sigma)$ where $\vec{\mu_q} = [\mu_1, \ldots, \mu_d]$ and $\Sigma_q = \mathrm{diag}(\sigma_1^2, \ldots, \sigma_d^2)$ be a normal distribution on $\mathbb{R}^d$.

We will compare using the following $q$ distributions in the definition of $\hat{p}_M$. We briefly describe them here and refer to Section (4.1) for the precise definitions.

- $q_{\text{IWAE}}$: draws $k$ independent samples from $q$.

- $q_{\text{IWAEN}}$: draws $k/2$ samples from $q$ and obtains the remaining $k/2$ samples by reflecting them through $\vec{\mu}$

- $q_{\text{COVK}}$: draws $k$ independent samples from $q$ and transforms them through multiplication by a matrix defined using as a kronecker product.

- $q_{\text{COVRn}}$: draws one sample from $q$ and rotates it around $\vec{\mu}$ to obtain $n - 1$ more samples. Repeats this $k/n$ times.

- $q_{\text{COVD}}$: draws one sample from $q$ and multiplies it by $k$ distinct matrices (restricted appropriately to ensure the marginals all equal $q$)

Additionally we use the following choices for $p$ as targets.

- Mixture: an equal mixture of two normal distributions

- Half: a standard normal with pdf scaled by $\epsilon = 0.01$ when $x_n > 0$ for a fixed coordinate $x_n$.

- Slice: a standard normal with pdf scaled by $\epsilon = 0.01$ when $x_i > 0$ for any coordinate $x_i$.

In the case of a VAE we train it on a binarized MNIST dataset according to the exact specifications of Burda et al. (2015). In this case the mean and standard deviation of $q$ as defined in this section are output by the encoder network once per datapoint, as in a usual VAE, but the remaing entries of the covariance matrix of the joint distribution do not depend on the data.

ACKNOWLEDGMENTS

Thanks for Chris Cremer and Will Grathwohl for helpful input in the early stages of the project, and on the application of these methods to VAEs.

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

## 4 APPENDIX

### 4.1 DETAILED MODEL DEFINITIONS

- $q_{\text{IWAE}}$: equal to $\prod_{i=1}^{k} q$

- $q_{\text{IWAEN}}$: the distribution on $\mathbb{R}^{dk}$ given by drawing $k/2$ samples $h_1, \ldots, h_{k/2}$ from $q$ and taking $k/2$ more samples to be $-h_1, \ldots, -h_{k/2}$.

Given any $nk \times nk$ matrix $\Sigma$ whose diagonal $d \times d$ blocks are all equal to $\Sigma_q$ we let $q_{\text{COV}}$ be the distribution on $\mathbb{R}^{dk}$ given by $N(\vec{\mu}, \Sigma)$ where $\vec{\mu} = [\vec{\mu_q}, \ldots, \vec{\mu_q}]$. We denote any sample $\vec{h} = [h_1, \ldots, h_k]$ with each $h_i \in \mathbb{R}^d$. We will test several versions of $q_{\text{COV}}$ with different matrices $\Sigma$.

- $q_{\text{COVKn}}$: let $L = (I_n \otimes B) \otimes A$ where $I_n \otimes B$ is of size $k \times k$ with ones on the diagonal and $A = \text{diag}(\sigma_1, \ldots, \sigma_d)$. Let $\Sigma = LL^T$ and let $q_{\text{COVKn}}$ denote the $q_{\text{COV}}$ with this $\Sigma$.

- $q_{\text{COVRn}}$: let $R$ be the $2 \times 2$ rotation matrix with angle $2\pi/n$. This matrix is fixed and not trainable. Let

$$B' = \begin{bmatrix} \Sigma_q^{1/2} \\ \vdots & & 0 \\ \Sigma_q^{1/2} \end{bmatrix} \begin{bmatrix} I \\ R \\ R^2 & & 0 \\ \vdots \\ R^{n-1} \end{bmatrix}$$

and let $B$ be the block diagonal matrix with $k/n$ diagonal blocks all of which are equal to $B'$. Then define $\Sigma = BB^T$. We let $q_{\text{COVRn}}$ denote $q_{\text{COV}}$ with this $\Sigma$.

- $q_{\text{COVD}}$: let

$$B = \begin{bmatrix} \Sigma_q^{1/2} \\ \vdots & & 0 \\ \Sigma_q^{1/2} \end{bmatrix} \begin{bmatrix} U_1 \\ U_2 & & 0 \\ \vdots \\ U_k \end{bmatrix}$$

where each $U_i$ has orthonormal rows and let $\Sigma = BB^T$. The entries of $U_i$ are trainable parameters. This generalizes the above distribution since $R$ is an orthonormal matrix.

### 4.2 ADDITIONAL REMARKS ABOUT EXPERIMENTS

One key difficulty in this approach is that it is not enough to learn a matrix of parameters to serve as $\Sigma$ since in general such a matrix will not be positive semi-definite as is required of a covariance matrix. We use the fact that for any invertible matrix $L$, $LL^T$ is positive-semidefinite, and instead learn this matrix.

Recall we are assuming $q = N(\mu, \Sigma)$. Sampling from $q$ can be performed by sampling $\epsilon \sim N(\vec{0}, I_{dk})$ and applying the transformation $\mu + L\epsilon$ where $L = \Sigma^{1/2}$ (that is $LL^T = \Sigma$). In principle the above remark implies that we should not lose anything by restricting to matrices $L$ where only the first $d$ columns are nonzero, since these corresponsd to using only the first sample, however in practice we found that learning a full matrix is beneficial.

### 4.3 AN EXAMPLE

We give a simple example where the optimal covariance matrix can be computed explicitly. In the case $d = 1$ and $k = 2$ let $q(h_1, h_2) = N(\vec{\mu}, \Sigma)$ where

$$\Sigma = \sigma^2 \cdot \begin{bmatrix} 1 & \rho \\ \rho & 1 \end{bmatrix}.$$

It is easy to see that $\text{Var}\hat{p}_M$ is minimized as $\rho \longrightarrow -1$. In this case $L = \sigma \begin{bmatrix} 1 & 0 \\ -1 & 0 \end{bmatrix}$. This results in a sampling scheme where the second sample is obtained from the first by reflecting it through the mean.

### 4.4 SOME OTHER ESTIMATORS

In the main text we only considered distributions $q(h_1, \ldots, h_k)$ where all the marginals are equal. Suppose the distribution $q$ in the definition of $\hat{p}_M$ is a product of $k$ distinct gaussians $q_i$ so that $q(h_i) = q_i(h_i)$. It is clear that we do not gain anything by introducing this addtitional generality as given any such $q$ we can construct a new distribution by setting all $q_i = q_j$ with $\text{Var}(p(x, h)/q_j(h))$ minimal while leaving the covariances the same.

Finally we remark on another natural choice of estimator. Define

$$\hat{p}_{AR}(h_1, \ldots, h_k) = \frac{1}{k} \sum_{i=1}^{k} \frac{p(x, h_i)}{q(h_i \mid h_1, \ldots h_{i-1})} \tag{4}$$

It is easy to see that this is also an unbiased estimator for $p(x)$.

Let $X_i = p(x, h_i)/q(h_i \mid h_1, \ldots h_{i-1})$ and $Y_i = p(x, h_i)/q(h_i)$ for all $i$. For simplicity we assume $k = 2$ in the following.

**Lemma 4.1.** *Let $k = 2$. Suppose $\text{Cov}(Y_1, Y_2) \leq 0$. Then $\text{Var}(\hat{p}_M) < \text{Var}(\hat{p}_{AR})$.*

*Proof.* We have

$$k^2 (\text{Var}(\hat{p}_{AR}) + p(x)^2) = \sum_i E(X_i^2) + 2E(X_1 X_2)$$

and similarly for $\hat{p}_M$.

Note $X_1 = Y_1$. For $i = 2$ we compute .

$$
\begin{aligned}
E(X_2^2) &= \sum_{h_1, h_2} q(h_1, h_2) \frac{p(x, h_2)^2}{q(h_2 \mid h_1)^2} \\
&= \sum_{h_1, h_2} \frac{q(h_1)}{q(h_2 \mid h_1)} p(x, h_2)^2 \\
&= \sum_{h_2} E_{q(h_1)} \left[ \frac{1}{q(h_2 \mid h_1)} \right] p(x, h_2)^2 \\
&\geq \sum_{h_2} \frac{1}{E_{q(h_1)} [q(h_2 \mid h_1)]} p(x, h_2)^2 \\
&= \sum_{h_2} q(h_2) \frac{p(x, h_2)^2}{q(h_2)^2} \\
&= E(Y_2^2).
\end{aligned}
$$

The inequality is obtained using $E[1/X] \geq 1/E[X]$ (with equality only if $\text{Var} X = 0$) for any strictly positive random variable $X$, which is a consequence of Jensen's inequality.

Now

$$
\begin{aligned}
E(X_1 X_2) &= \sum_{h_1, h_2} q(h_1, h_2) \frac{p(x, h_1) p(x, h_2)}{q(h_1) q(h_2 \mid h_1)} \\
&= p(x)^2.
\end{aligned}
$$

Hence $\text{Cov}(X_1, X_2) = 0$. $\qquad\qquad\square$

### 4.5 THE VARIANCE OF $\hat{p}_{EW}$

Let $q(h)$ be any distribution on $Z$. Define $w_i = p(x, h_i)/q(h_i)$ and

$$\tilde{q}(h_k) = E_{q(h_1, \ldots h_{k-1})} \left[ \frac{p(x, h_k)}{\frac{1}{k} \sum w_i} \right].$$

In Cremer et al. (2017) they define the estimator

$$\hat{p}_{EW}(h_k) = \frac{p(x, h_k)}{\tilde{q}(h_k)}$$

and show it yields a strictly better variational lower bound of $p(x)$. Here we will show $\text{Var}(\hat{p}_{EW}) < \text{Var}(\hat{p}_{IWAE})$.

**Lemma 4.2.** $\text{Var}(\hat{p}_{EW}) < \text{Var}(\hat{p}_{IW})$.

*Proof.* Let $X = \hat{p}_{EW}$. We again use the fact $E[1/X] \geq 1/E[X]$ to compute

$$E(X^2) = \sum_{h_k} \tilde{q}(h_k) \frac{p(x, h_k)^2}{\tilde{q}(h_k)^2}$$

$$= \sum_{h_k} \frac{1}{E_{q(h_1,...h_{k-1})}\left[\frac{1}{\frac{1}{k}\sum w_i}\right]} p(x, h_k)$$

$$\leq \sum_{h_k} E_{q(h_1,...h_{k-1})}\left[\frac{1}{k}\sum_i w_i\right] p(x, h_k)$$

$$= \frac{1}{k}\sum_{h_k}(\sum_i \sum_{h_i} p(x, h_i)p(x, h_k)) + \frac{p(x, h_k)^2}{q(h_k)}$$

$$= \frac{k-1}{k}p(x)^2 + \frac{1}{k}E_{q(h_k)}((p(x, h_k)/q(h_k))^2)$$

and the inequality is not strict only if $\text{Var}_{q(h_1,...h_{k-1})}(\frac{1}{k}\sum w_i) = 0$. Thus

$$\text{Var}(X) < \frac{1}{k}\text{Var}(p(x, h)/q(h))$$

$$= \text{Var}(\hat{p}_{IW}).$$

$\square$

### 4.6 PROOF OF THEOREM (2.1)

*Proof.* Let $\epsilon > 0$. Let $U_1 \subset Z$ be the set on which $|\hat{\theta} - \theta| < \epsilon$ and $U_2 = Z \backslash U_1$. By Chebyshev's inequality $\mu(U_2) < \text{Var}(\hat{\theta})/\epsilon^2$. Let

$$\Delta = \frac{\hat{\theta}_i - \theta}{\theta}.$$

We have

$$\log\theta - \mathcal{L}(\hat{\theta}) = \int_Z \log\theta - \log\hat{\theta}d\mu$$

$$= \int_Z \log(\Delta + 1)d\mu$$

$$= \int_{U_1} \log(\Delta + 1)d\mu + \int_{U_2} \log(\Delta + 1)d\mu.$$

The Taylor expansion of $\log(1 + \Delta)$ is

$$\Delta - \frac{\Delta^2}{2} + \frac{\Delta^3}{3} - \cdots.$$

Then

$$\int_{U_1} \log(\Delta + 1)d\mu < \epsilon - \frac{\epsilon^2}{2}\cdots < C_1\epsilon$$

for some constant $C_1$. Furthermore $\int_Z \log(\Delta + 1)d\mu < C_2$ (for instance $C_2 = \log p(x)$) so it follows that

$$\int_{U_2} \log(\Delta + 1)d\mu < C_2 \frac{\mathrm{Var}(\hat{\theta})}{\epsilon^2}.$$

Thus we have shown

$$\log\theta - \mathcal{L}(\hat{\theta}_i) < C_1\epsilon + C_2 \frac{\mathrm{Var}(\hat{\theta})}{\epsilon^2}.$$

$\square$

Let $\mathcal{E} = \log\theta - \mathcal{L}(\hat{\theta}_i)$. Proposition 1 in Maddison et al. (2017) bounds the difference between $\mathcal{E}$ and $\mathrm{Var}(\hat{\theta})$ in terms of the 6th central moment of $\hat{\theta}$ (and under finiteness of the first inverse moment of $\hat{\theta}$), which does not imply that $\mathcal{E} \longrightarrow 0$ as $\mathrm{Var}(\hat{\theta}) \longrightarrow 0$. However in situations where these quantities are known their result provides a more precise relationship between $\mathcal{E}$ and $\mathrm{Var}(\hat{\theta})$.

