# OpenReview forum: "Joint Importance Sampling for Variational Inference"
_ICLR.cc/2018/Workshop — Reject_

### Official Review · AnonReviewer1 · 2018-03-09
**Interesting**

**Rating:** 5
**Confidence:** 4

**Review:**

The authors describe ways of reducing the variance of importance sampling estimates. A by-product is that evidence lower bounds obtained from IS estimates with lower variance are typically tighter.
- several ideas are indeed well known. For example, q_{IWAEN} rely on the standard idea of antithetic variables. q_{COVRn} can be seen as an approximate Rao-blackwellization, etc... : in general, if a distribution is invariant under the applications of a transformation T, then one can indeed consider estimators of the type (phi(X) + phi(T(x)) + phi(T^2(x)) + ... + phi(T^n(X)) )/n.
- numerical simulations are not terribly exciting. The most complicated setting is VAE on MNIST, and in this case the proposed methods are not performing better than standard IWAE. Maybe surprisingly, estimators with trainable parameters such as q_{COVD} do not appear to perform better on MNIST?

If one could easily and computationally cheaply reduce the variability of IS estimates, that would indeed be fantastic. The paper is interesting, but after reading it, I still feel that no real/novel solution has been proposed.

---

### Official Review · AnonReviewer3 · 2018-03-16
**Provides an interesting theorem**

**Rating:** 6
**Confidence:** 3

**Review:**

The basic idea behind this paper is interesting and to the best of my knowledge new. I did not carefully check the Lemmata 4.1 and 4.2; which are statements about specific importance sampling estimators the authors use in the experimental section. But I went through the derivation of Theorem 2.1. Considering the public comments and replies by the authors, I believe the proof and the main statements are correct (once the authors update the paper). That theorem is also general enough that it might play a role in future research in this direction. On the other hand: the experiments could not convince me that there is much practical value in the proposed idea (for now); the manuscript overall seems a bit rushed with some mathematical typos discussed in the public forum.

Assuming the typos and the proof will be corrected, I think this could be a reasonable workshop contribution which might be picked up in future research.

---

### Public Comment · ~Christian_A_Naesseth1 · 2018-02-26
**Some comments**

Seems like an interesting idea, learning correlation between the importance samples to try to lower the variance (and tighten the bound) of the normalization constant estimate. I would appreciate if you added a citation to my Variational Sequential Monte Carlo paper, it is highly relevant to the discussion of q_EW/\tilde q in section 2.1 and the appendix.

I think there might be some mistakes in Thm 2.1 and the proofs. The definition of delta does not align with the replacement you do in section 4.6, there is a minus sign missing after the second equality in the equation following "We have ..." It was also not clear to me how you bound the integral using C_2=log p(x)? That integral is exactly the difference log p(x) - E[log \hat p(x)], and using C_2=log p(x) gives you the claim that E[log\hat p(x)] > 0 which I do not think is true in general.

Some other things I saw while reading:
- Seems there is some inconsistency in the notation between h_i and Y_i, one example is above eq. (3)
- Below Thm 2.1 you are missing "\log" before "\theta" in two places
- Last paragraph in section 2.1, minimizes KL(q_EW||p) and not KL(p||q_EW).

---

> ### Public Comment · ~Jack_Klys1 · 2018-03-04
> **response**
>
> Thanks for your comment.
> You are right in both your points about Thm 2.1.
> The first point doesn't affect the proof but the second one does. However with minor modifications the proof still works to show that the bias log theta - E[log \hat theta_i] goes to 0 as the variance Var[\hat theta_i] goes to 0 under the additional assumption that the sequence {log \hat theta_i} is uniformly integrable (also assumed in Maddison et al. whose proof we follow partially).
> So the statement of Thm 2.1 should be changed to just say the above.
> The other points will also be corrected including the citation.

---

> > ### Public Comment · ~Christian_A_Naesseth1 · 2018-03-05
> > **thanks**
> >
> > Hi Jack,
> >
> > Yes, with the uniform integrability assumption the result should follow without a hitch. I think another way of proving it is using the fact that \hat \theta_i -> \theta_i, because unbiased and Var(\hat\theta_i) -> 0. Then use continuous mapping theorem for -\log(\hat\theta_i).
> >
> > Thanks!

---

### Decision · Program_Chairs · 2018-03-20
**ICLR 2018 Workshop Acceptance Decision**

**Decision:**

Reject

**Comment:**

Based on the reviews, this paper has not been accepted for presentation at the ICLR workshop. However, the conversation and updates can continue to appear here on OpenReview.